# Assessing pregnancy and neonatal outcomes in Malawi, South Africa, Uganda, and Zimbabwe: Results from a systematic chart review

**Jennifer E. Balkus**[1,2]*, **Moni Neradilek**[2], **Lee Fairlie**[3], **Bonus Makanani**[4], **Nyaradzo Mgodi**[5], **Felix Mhlanga**[5], **Clemensia Nakabiito**[6], **Ashley Mayo**[7], **Tanya Harrell**[2], **Jeanna Piper**[8], **Katherine E. Bunge**[9], on behalf of the MTN-042B Study Team¶

1 Department of Epidemiology, University of Washington, Seattle, Washington, United States of America,
2 Vaccine and Infectious Diseases Division, Fred Hutchinson Cancer Research Center, Seattle, Washington, United States of America, 3 Faculty of Health Sciences, Wits Reproductive Health and HIV Institute, University of the Witwatersrand, Gauteng, South Africa, 4 College of Medicine-Johns Hopkins Research Project, Blantyre, Malawi, 5 University of Zimbabwe College of Health Sciences Clinical Trials Research Centre, Harare, Zimbabwe, 6 Makerere University-Johns Hopkins University Research Collaboration, Kampala, Uganda, 7 FHI 360, Durham, North Carolina, United Stated of America, 8 US National Institutes of Health, Bethesda, Maryland, United States of America, 9 Department of Obstetrics and Gynecology, University of Pittsburgh, Pittsburgh, Pennsylvania, United Stated of America

¶ The complete authorship list can be found in the Acknowledgements
* jbalkus@uw.edu

**Data Availability Statement:** All relevant data are within the manuscript and its Supporting information files.

## Abstract

A systematic chart review was performed to estimate the frequency of pregnancy outcomes, pregnancy complications and neonatal outcomes at facilities in Blantyre, Malawi; Johannesburg, South Africa; Kampala, Uganda; and Chitungwiza and Harare, Zimbabwe to provide comparisons with estimates from an ongoing clinical trial evaluating the safety of two biomedical HIV prevention interventions in pregnancy. A multi-site, cross-sectional chart review was conducted at Maternal Obstetric Units and hospitals where women participating in the ongoing clinical trial would be expected to deliver. All individuals delivering at the designated facilities or admitted for postpartum care within seven days of a delivery elsewhere (home, health clinic, etc.) were included in the review. Data were abstracted for pregnancy outcomes, pregnancy complications, maternal and neonatal death, and congenital anomalies. Data from 10,138 records were abstracted across all four sites (Blantyre n = 2,384; Johannesburg n = 1,888; Kampala n = 3,708; Chitungwiza and Harare n = 2,158), which included 10,426 pregnancy outcomes. The prevalence of preterm birth was 13% (range across sites: 10.4–20.7) and 4.1% of deliveries resulted in stillbirth (range: 3.1–5.5). The most commonly noted pregnancy complication was gestational hypertension, reported among 4.4% of pregnancies. Among pregnancies resulting in a live birth, 15.5% were low birthweight (range: 13.8–17.4) and 2.0% resulted in neonatal death (range:1.2–3.2). Suspected congenital anomalies were noted in 1.2% of pregnancies. This study provides systematically collected data on background rates of pregnancy outcomes, pregnancy complications and neonatal outcomes that can be used as a reference in support of ongoing HIV prevention studies. In

**Funding:** The Microbicide Trials Network is funded by the National Institute of Allergy and Infectious Diseases (UM1AI068633, UM1AI068615, UM1AI106707), with co-funding from the Eunice Kennedy Shriver National Institute of Child Health and Human Development and the National Institute of Mental Health, all components of the U.S. National Institutes of Health (NIH). The content is solely the responsibility of the authors and does not necessarily represent the official views of the NIH. The funders had no role in study design, data collection and analysis, decision to publish, or preparation of the manuscript.

**Competing interests:** The authors have declared that no competing interests exist.

addition, estimates from this study provide important background data for future studies of investigational products evaluated in pregnancy in these urban settings.

## Introduction

Globally, reproductive-aged cisgender women continue to experience the largest number of new HIV infections [1]. Pregnancy represents a period of increased HIV susceptibility, with higher rates of HIV acquisition compared to non-pregnant women [2]. As novel HIV prevention products are developed and demonstrate reductions in HIV incidence [3–5], it is critical that these investigational products be carefully evaluated during pregnancy to ensure their safety and effectiveness for use during this complex time [6]. In support of this imperative, the Microbicide Trials Network (MTN) is conducting the MTN-042/DELIVER study—a phase IIIb, randomized, open-label safety trial of the dapivirine vaginal ring (25 mg) and oral pre-exposure prophylaxis (PrEP) (200 mg emtricitabine [FTC]/300 mg tenofovir disoproxil fumarate [TDF]) for HIV prevention in cisgender pregnant women (ClinicalTrials.gov #NCT03965923) [7]. The primary objectives of the DELIVER study are to describe maternal and neonatal safety and pregnancy outcomes among individuals randomized to receive the dapivirine vaginal ring or oral PrEP. Since all pregnant individuals enrolled will use an HIV prevention product, the frequency of pregnancy outcomes, pregnancy complications and neonatal outcomes will be compared to rates in the general population where the DELIVER study is being conducted.

While national data are generally available for key maternal and neonatal health indicators, such as maternal mortality, neonatal mortality and low birth weight, systematically collected data for other important outcomes and pregnancy-related complications, which may be impacted by investigational products, are sparse [8–10]. To address this gap, we performed a systematic chart review to estimate the frequency of pregnancy outcomes, pregnancy complications and neonatal outcomes at facilities in Blantyre, Malawi; Johannesburg, South Africa; Kampala, Uganda; and Chitungwiza and Harare, Zimbabwe. The primary aim of this study was to describe the frequency of pregnancy outcomes, pregnancy complications and neonatal outcomes among individuals delivering in specified catchment areas and compare those with estimates from an ongoing clinical trial evaluating the safety of the two biomedical interventions for HIV prevention in pregnancy [7]. These data will be used both to support the DELIVER study and provide important background data for future studies of investigational products evaluated in pregnancy.

## Materials and methods

The following institutional review boards (IRB) or ethics committees (EC) reviewed this research: University of the Witwatersrand's Human Research Ethics Committee (Johannesburg, South Africa site); Medical Research Council of Zimbabwe Institutional Review Board (Harare, Zimbabwe site); University of Malawi College of Medicine Research Ethics Committee and Johns Hopkins School of Public Health Institutional Review Board (Blantyre, Malawi site); Joint Clinical Research Center IRB/EC, Johns Hopkins Medicine Office of Human Subjects Research IRB, and Uganda National Council for Science and Technology (Kampala, Uganda site). Each site received approval and was granted a waiver of informed consent.

A multi-site, cross-sectional chart review was conducted at hospitals within proximity to clinical research sites in Blantyre, Malawi, Johannesburg, South Africa, Kampala, Uganda, and

Chitungwiza and Harare, Zimbabwe [11]. Initially, each site identified primary and/or tertiary facilities that would serve the catchment area surrounding the research clinic in order to assure that all deliveries from their catchment area were represented. The maternity care system in each of these settings is based upon a group of primary care facilities that feed into the tertiary care facility for their health region, thus all pregnant individuals in the catchment area would be expected to deliver in one of these two locations. Pregnant individuals typically present to the primary care facility closest to their home but are transferred to the tertiary facility if they are either known to be high risk upon presentation from their antenatal care (e.g., prior cesarean delivery) or develop high risk characteristics during labor, delivery or postpartum (e.g., preeclampsia). Both low and high-risk individuals may also present directly to the tertiary facility, or in some cases, the tertiary facility also serves as the primary facility for the catchment area around the research clinic. At the time data abstraction was initiated in Zimbabwe, there was a localized industrial action (protests) by health care workers which affected operations at the referral hospital. As a result, at study initiation the number of deliveries was lower than expected as patients in the catchment area chose to deliver at alternative clinics. Data abstraction was temporarily paused in Zimbabwe and additional facilities were added to reach a sufficient number of deliveries. Details on the timing of data abstraction and facilities are provided in Table 1.

At each facility, medical records were identified through birth registries, maternal case records, delivery records, or admission logs over an eight-week period. All individuals delivering at the designated facility or admitted to the facility for postpartum care within seven days of a delivery elsewhere (home, health clinic, etc.) were included in the review. Patient charts identified for abstraction were assigned a unique record number. Study staff all had prior training in healthcare (medicine, nursing, or other medical background) and received study specific training on chart review and data abstraction for this project. Records were abstracted by study staff within seven days of the date of the pregnancy outcome using a standardized electronic data abstraction form. Study staff had no interaction with individuals who gave birth or their health care providers. Data were abstracted for pregnancy outcomes (full term birth, preterm birth, stillbirth or fetal demise), pregnancy complications (hypertensive disorders, post-partum hemorrhage, fever of unclear etiology, chorioamnionitis, and endometritis) and infant outcomes (low birth weight and suspected congenital anomalies) based on the presence of the diagnosis term in the chart or presumed presence of the condition based on additional data present in the chart. Estimated gestational age at delivery was assessed using a combination of data from the medical chart (as available), including date of last menstrual period, estimated date of delivery based on ultrasound and fundal height. Maternal and neonatal deaths were also assessed. Definitions for pregnancy related complications are provided in Table 2 and the data abstraction form is provided in S1 File.

Data were entered using web-based REDCap (Nashville, TN, USA) or REDCap mobile, which allows for direct data entry on mobile devices (smart phones, tablets) [12]. If technology issues emerged, data abstraction was shifted to paper case report forms and entered into the database once the technology issues were resolved. Routine data and logic checks were built into the database to ensure data quality. As an additional quality control check, each day approximately 10% of records at each facility were randomly selected for a second review. Abstraction results were compared, any discrepancies were discussed and after the reviewers reached agreement, the data were updated as needed. Descriptive statistics were utilized to summarize the frequency of the pregnancy outcomes, pregnancy complications and neonatal outcomes overall and by site. A high proportion of records reviewed indicated that the patient had been transferred to the current facility for care; therefore, a post-hoc exploratory analysis was performed using descriptive statistics to summarize select outcomes by transfer status.

**Table 1. Data abstraction locations and dates by site.**

| City, Country | Abstraction location | Primary or tertiary facility | Dates | Number of records |
|---|---|---|---|---|
| Johannesburg, South Africa | Charlotte Maxeke Johannesburg Academic Hospital[1] | Tertiary | 01-Oct-2019–07 Oct-2019; | 1455 |
| | | | 15-Oct-2019–02-Dec-2019 | |
| | Shandukani Maternal and Obstetrics Unit | Primary | 05-Nov-2019–30-Dec-2019 | 433 |
| Blantyre, Malawi | Queen Elizabeth Central Hospital | Tertiary | 16-Aug-2019–10-Oct-2019 | 2016 |
| | Ndirande Health Centre | Primary | 16-Aug-2019–10-Oct-2019 | 368 |
| Kampala, Uganda | Kawempe General Hospital | Primary and Tertiary[4] | 18-Sep-2019–12-Nov-2019 | 3708 |
| Chitungwiza and Harare, Zimbabwe | Zengeza Municipal Clinic[1] | Primary | 29-Sep-2019–03-Oct-2019; | 177 |
| | | | 12-Oct-2019–29-Nov-2019 | |
| | Mbuya Nehanda Maternity Hospital, Parirenyatwa Hospital Group[1,2] | Tertiary | 28-Sep-2019–04-Oct-2019; | 578 |
| | | | 12-Oct-2019–28-Oct-2019; | |
| | | | 07-Dec-2019–06-Jan-2020 | |
| | Chitungwiza Central Hospital[3] | Tertiary | 10-Feb-2020–18-Mar-2020 | 819 |
| | Edith Opperman Maternity Clinic[3] | Primary | 10-Feb-2020–18-Mar-2020 | 584 |

[1]Some clinics underwent a one-week pause after completing the first week of data collection to allow for the study team to review data ensure the data were recorded as expected.

[2]During the time of the abstraction, industrial action (protests) occurred at Mbuya Nehanda Maternity Hospital, Parirenyatwa Group of Hospitals.

[3]Additional facilities added to close gap in reduced number of charts due to protests.

[4] This facility serves as both a primary facility for uncomplicated deliveries and also a provides care for individuals referred from other settings (tertiary facility).

Data from prior studies suggest differences in the pregnancy outcomes by maternal HIV status [13,14]; therefore, a sensitivity analysis was performed assessing outcomes stratified by maternal HIV status and outcome frequencies were compared using Chi-squared tests at alpha = 0.05. All analyses were conducted using R, version 3.5.3 (R Core Team, Vienna, Austria).

## Results and discussion

Data from 10,138 records were abstracted across all four clinical research sites (Blantyre, Malawi = 2,384; Johannesburg, South Africa = 1,888; Kampala, Uganda = 3,708; Chitungwiza and Harare, Zimbabwe = 2,158), which included 10,426 pregnancy outcomes. A detailed summary of maternal characteristics for the entire cohort and by site are presented in Table 3. Overall, mean maternal age was 26.4 years, mean gravidity was 2.5, 43.1% were transferred to the current facility for care and 13.6% were living with HIV. Attendance at four or more antenatal visits varied from 33.9% in Chitungwiza and Harare to 70.2% in Johannesburg. The vast majority of pregnancies (97.2%) were singleton, had deliveries that occurred at the health

**Table 2. Definitions of pregnancy-related complications.**

| Term | Study Definition |
| --- | --- |
| *Hypertension* | |
| Gestational | Pregnancy >20 weeks and new diagnosis of hypertension (>140 mmHg systolic and/or > 90 mmHg) without severe features of pre-eclampsia or proteinuria |
| Pre-eclampsia without severe features | Pregnancy >20 weeks and new diagnosis of hypertension (≥140 mmHg systolic and/or ≥ 90mmHg) and proteinuria but no severe features, which include:<br>• Severely elevated blood pressures, with systolic blood pressure ≥160 mmHg and/or diastolic blood pressure ≥110 mmHg, which is confirmed after only minutes (to facilitate timely antihypertensive treatment)<br>• Development of a severe headache (which can be diffuse, frontal, temporal or occipital) that generally does not improve with over the counter pain medications (such as acetaminophen/paracetamol)<br>• Development of visual changes (including photopsia, scotomata, cortical blindness)<br>• Eclampsia, or new-onset grand mal seizures in a patient with preeclampsia, without other provoking factors (such as evidence of cerebral malaria or preexisting seizure disorder). Seizures are often preceded by headaches, visual changes or altered mental status<br>• New onset thrombocytopenia, with platelet count <100,000/μL<br>• New onset of nausea, vomiting, epigastric pain<br>• Transaminitis (AST and ALT elevated to twice the upper limit of normal)<br>• Liver capsular hemorrhage or liver rupture<br>• Worsening renal function, as evidenced by serum creatinine level greater than 1.1 mg/dL or a doubling of the serum creatinine (absent other renal disease)<br>• Oliguria (urine output <500 mL/24 h)<br>• Pulmonary edema (confirmed on clinical exam or imaging) |
| Pre-eclampsia with severe features | Pregnancy >20 weeks and new diagnosis of hypertension (≥140 mmHg systolic and/or ≥ 90mmHg) and proteinuria with severe features (see list above) |
| Eclampsia | Based on diagnosis term in chart |
| *Other complications* | |
| Postpartum Hemorrhage | Based on diagnosis term in chart |
| Fever of Unclear Etiology | Based on diagnosis term in chart or report of mother with a temperature >38.5 degrees Celsius |
| Chorioamnionitis | Based on diagnosis term in chart or report of mother with a temperature >38 degrees Celsius and treated with antibiotics during labor |
| Postpartum Endometritis | Based on diagnosis term in chart or report of mother with a temperature >38 degrees Celsius after delivery and treated with antibiotics |

facility where data abstraction was performed (93.8%) and resulted in vaginal deliveries (73.1%) (Table 4).

Data on pregnancy outcomes were available for 10,180 (97.6%) pregnancies (Table 4). Among records with complete data on pregnancy outcomes, 413 (4.1%) resulted in a stillbirth or fetal demise (range: 3.1–5.5) and 1,319 (13.0%) resulted in preterm deliveries (range: 10.7–21.6) (Fig 1). Among pregnancies resulting in a livebirth, 1,539 (15.5%) were low birthweight (range: 13.8–17.4) and 2.0% resulted in neonatal deaths (range: 1.2–3.2). Suspected congenital anomalies were noted in 125 (1.2%) charts (Table 5), with the most common being polydactyly (n = 47), followed by musculoskeletal anomalies (n = 24). Patterns of reported anomalies appeared similar across sites.

Maternal outcomes and complications are presented in Table 6. Maternal death was rare and only reported by the Blantyre site (7 deaths (0.29%)). The most common complication was gestational hypertension, which was reported among 4.4% of pregnancies and varied by site with the lowest prevalence in Kampala, Uganda (0.9%) and the highest prevalence in

**Table 3. Maternal characteristics at data abstraction overall and by site.**

| | Malawi | Uganda | South Africa | Zimbabwe | Overall |
|---|---|---|---|---|---|
| | N = 2384 | N = 3708 | N = 1888 | N = 2158 | N = 10138 |
| Maternal age | 25.7±6.7 (14–48) | 25.5±5.6 (13–46) | 28.6±6.1 (12–48) | 26.7±6.4 (14–45) | 26.4±6.3 (12–48) |
| Gravidity | 2.5±1.6 (1–10) | 2.6±1.7 (0–11) | 2.6±1.3 (1–12) | 2.4±1.3 (1–8) | 2.5±1.5 (0–12) |
| Parity | 1.4±1.5 (0–9) | 1.4±1.6 (0–9) | 1.3±1.1 (0–6) | 1.3±1.3 (0–7) | 1.4±1.4 (0–9) |
| Attended 4+ antenatal care visits | 1070 (46.7%) | 578 (38.9%) | 1259 (70.2%) | 633 (33.9%) | 3540 (47.6%) |
| HIV status | | | | | |
| Negative | 2008 (84.2%) | 3185 (85.9%) | 1409 (74.6%) | 1765 (81.8%) | 8367 (82.5%) |
| Positive | 311 (13.0%) | 391 (10.5%) | 463 (24.5%) | 216 (10.0%) | 1381 (13.6%) |
| Documented as unknown | 22 (0.9%) | 12 (0.3%) | 2 (0.1%) | 100 (4.6%) | 136 (1.3%) |
| Not documented | 43 (1.8%) | 120 (3.2%) | 14 (0.7%) | 77 (3.6%) | 254 (2.5%) |
| Transferred for delivery from a different facility | 1681 (70.5%) | 1164 (31.4%) | 741 (39.9%) | 756 (35.5%) | 4342 (43.1%) |
| Transferred to a different facility after delivery | 2 (0.1%) | 8 (0.2%) | 13 (0.7%) | 32 (1.5%) | 55 (0.5%) |
| Number of infants from this delivery | | | | | |
| 1 | 2327 (97.6%) | 3584 (96.7%) | 1843 (97.6%) | 2103 (97.5%) | 9857 (97.2%) |
| 2 | 56 (2.3%) | 121 (3.3%) | 42 (2.2%) | 55 (2.5%) | 274 (2.7%) |
| 3 | 1 (0.0%) | 3 (0.1%) | 3 (0.2%) | 0 (0.0%) | 7 (0.1%) |

Statistics are N (%) for categorical variables and mean±SD (range) for continuous variables.

**Table 4. Pregnancy outcomes and characteristics at delivery.**

| | Malawi | Uganda | South Africa | Zimbabwe | Overall |
|---|---|---|---|---|---|
| | N = 2442 | N = 3835 | N = 1936 | N = 2213 | N = 10426 |
| Place of delivery | | | | | |
| Current health facility | 2282 (93.4) | 3607 (94.1) | 1858 (96.0) | 2032 (91.8) | 9779 (93.8) |
| At a different health facility | 131 (5.4) | 180 (4.7) | 26 (1.3) | 71 (3.2) | 408 (3.9) |
| At a home (private residence) | 28 (1.1) | 33 (0.9) | 48 (2.5) | 109 (4.9) | 218 (2.1) |
| Not documented | 1 (0.0) | 15 (0.4) | 4 (0.2) | 1 (0.0) | 21 (0.2) |
| Mode of delivery | | | | | |
| Vaginal delivery | 1819 (74.5) | 2726 (71.1) | 1258 (65.0) | 1819 (82.2) | 7622 (73.1) |
| Cesarean delivery | 592 (24.2) | 1082 (28.2) | 674 (34.8) | 362 (16.4) | 2710 (26.0) |
| Other | 28 (1.1) | 9 (0.2) | 0 (0.0) | 32 (1.4) | 69 (0.7) |
| Not documented | 3 (0.1) | 18 (0.5) | 4 (0.2) | 0 (0.0) | 25 (0.2) |
| Pregnancy outcome | | | | | |
| Full term live birth | 2094 (85.7) | 3192 (83.2) | 1387 (71.6) | 1775 (80.2) | 8448 (81.0) |
| Premature live birth (<37 weeks) | 261 (10.7) | 400 (10.4) | 400 (20.7) | 258(11.7) | 1319 (12.7) |
| Still born/intrauterine fetal demise (≥20 weeks) | 75 (3.1) | 152 (4.0) | 64 (3.3) | 122 (5.5) | 413 (4.0) |
| Outcome not documented | 12 (0.5) | 91 (2.4) | 85 (4.4) | 58 (2.6) | 246 (2.4) |
| Birthweight (grams)[2] | 2934±533 (600–5000) | 3027±633 (500–5400) | 2946±659 (500–5570) | 2977±592 (400–4960) | 2979±608 (400–5570) |
| Low birthweight (<2500 g)[2] | 353 (15.1) | 574 (15.9) | 324 (17.4) | 288 (13.8) | 1539 (15.5) |
| Neonatal death[3] | 61 (2.6) | 42 (1.2) | 24 (1.3) | 65 (3.2) | 192 (2.0) |

[1]Statistics are N (%) for categorical variables and mean ± SD (range) for continuous variables.

[2]Birthweight reported for livebirths only.

[3]Livebirths only. Missing data for N = 118 (1.2%) included in the denominator.

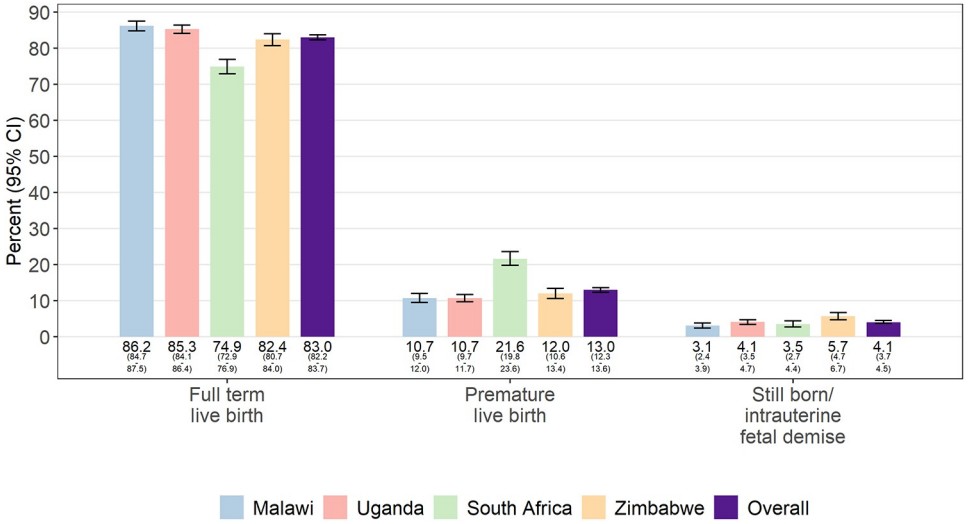

**Fig 1. Frequency of pregnancy outcomes by site.** Data on pregnancy outcomes, overall and by site, for n = 10,180 pregnancy outcomes. Records where the pregnancy outcome could not be ascertained were excluded.

Chitungwiza and Harare, Zimbabwe (9.3%) (Fig 2). Pre-eclampsia without and with severe features was noted among 2.2% and 2.1% of pregnancies, respectively, while eclampsia was recorded in 63 (0.6%) records. Postpartum hemorrhage was the next most common complication and was reported in 3.2% of pregnancies (range: 2.5–5.0). All other complications, including fever of unclear etiology, chorioamnionitis and postpartum endometritis, occurred in less than 1% of records reviewed.

Given the high proportion of records indicating a transfer from another facility for delivery (42.1%), we conducted a post-hoc exploratory analysis to summarize the frequency of select outcomes by transfer status. Frequencies were qualitatively similar when comparing those who were transferred from another facility for delivery and those who were not, with a slightly higher proportion of records reporting cesarean delivery (32.4% versus 21.0%) or preterm

**Table 5. Frequency of select suspected infant congenital anomalies identified at delivery[1].**

| | Malawi | | Uganda | | South Africa | | Zimbabwe | | Overall | |
|---|---|---|---|---|---|---|---|---|---|---|
| | N = 2442 | | N = 3835 | | N = 1936 | | N = 2213 | | N = 10426 | |
| | n | (%) | n | (%) | n | (%) | n | (%) | n | (%) |
| Any congenital anomaly identified at delivery[2] | 33 | (1.4) | 19 | (0.5) | 41 | (2.1) | 32 | (1.4) | 125 | (1.2) |
| Polydactyly | 16 | (0.7) | 3 | (0.1) | 11 | (0.6) | 17 | (0.8) | 47 | (0.5) |
| Musculoskeletal including clubfoot | 7 | (0.3) | 2 | (0.1) | 10 | (0.5) | 5 | (0.2) | 24 | (0.2) |
| Cleft Lip and/or Palate | 2 | (0.1) | 5 | (0.1) | 3 | (0.2) | 1 | (0.0) | 11 | (0.1) |
| Neural tube defects and/or Hydrocephalus | 3 | (0.1) | 5 | (0.1) | 2 | (0.1) | 0 | (0.0) | 10 | (0.1) |
| Umbilical Hernia | 3 | (0.1) | 1 | (0.0) | 4 | (0.2) | 2 | (0.1) | 10 | (0.1) |
| Esophageal, gastrointestinal, or anorectal | 2 | (0.1) | 1 | (0.0) | 1 | (0.1) | 2 | (0.1) | 6 | (0.1) |
| Genitourinary | 1 | (0.0) | 1 | (0.0) | 2 | (0.1) | 1 | (0.0) | 5 | (0.0) |
| Trisomies | 0 | (0.0) | 1 | (0.0) | 1 | (0.1) | 1 | (0.0) | 3 | (0.0) |

[1]Missing data for N = 120 (1.2%) included in the total.

[2]Suspective congenital anomalies occurring in >1 deliveries are included in the table by name. All suspected anomalies are included in the "Any congenital anomaly" category.

**Table 6. Frequency of maternal deaths and pregnancy complications.**

|  | Malawi | | Uganda | | South Africa | | Zimbabwe | | Overall | |
|---|---|---|---|---|---|---|---|---|---|---|
|  | N = 2384 | | N = 3708 | | N = 1888 | | N = 2158 | | N = 10138 | |
|  | n | (%) | n | (%) | n | (%) | n | (%) | n | (%) |
| Maternal death | 7 | (0.29) | 0 | (0) | 0 | (0) | 0 | (0) | 0 | (0) |
| Hypertension[1] |  |  |  |  |  |  |  |  |  |  |
| Any hypertension | 320 | (13.4) | 234 | (6.3) | 250 | (13.2) | 275 | (12.7) | 1079 | (10.6) |
| Chronic | 16 | (0.7) | 2 | (0.1) | 51 | (2.7) | 10 | (0.5) | 79 | (0.8) |
| Gestational | 128 | (5.4) | 33 | (0.9) | 88 | (4.7) | 200 | (9.3) | 449 | (4.4) |
| Pre-eclampsia WITHOUT severe features | 60 | (2.5) | 89 | (2.4) | 47 | (2.5) | 24 | (1.1) | 220 | (2.2) |
| Pre-eclampsia WITH severe features | 57 | (2.4) | 94 | (2.5) | 50 | (2.6) | 16 | (0.7) | 217 | (2.1) |
| Eclampsia | 18 | (0.8) | 13 | (0.4) | 8 | (0.4) | 24 | (1.1) | 63 | (0.6) |
| Not specified | 41 | (1.7) | 3 | (0.1) | 6 | (0.3) | 2 | (0.1) | 52 | (0.5) |
| Other complications[1] |  |  |  |  |  |  |  |  |  |  |
| Postpartum Hemorrhage | 120 | (5.0) | 105 | (2.8) | 49 | (2.6) | 54 | (2.5) | 328 | (3.2) |
| Fever of Unclear Etiology | 4 | (0.2) | 3 | (0.1) | 2 | (0.1) | 4 | (0.2) | 13 | (0.1) |
| Chorioamnionitis | 2 | (0.1) | 13 | (0.4) | 6 | (0.3) | 2 | (0.1) | 23 | (0.2) |
| Postpartum Endometritis | 27 | (1.1) | 3 | (0.1) | 3 | (0.2) | 7 | (0.3) | 40 | (0.4) |

[1]Missing data was <1% for hypertension outcomes and other complications. Records with missing data were included in the denominator.

birth (14.6% versus 10.9%) among those transferred to a different facility for delivery versus those that not been transferred, respectively (Table 7).

Among pregnancies where maternal HIV status was documented (n = 10,017; 96.1%), the frequency of preterm birth and stillbirth were lower among individuals who were HIV-negative compared to those living with HIV (Fig 3; p<0.001 and 0.2, respectively). With regard to pregnancy complications, hypertension-related complications and chorioamnionitis were more frequently documented among individuals who were HIV-negative (p<0.001 and p = 0.03, respectively), while other complications did not differ by HIV status (Fig 3).

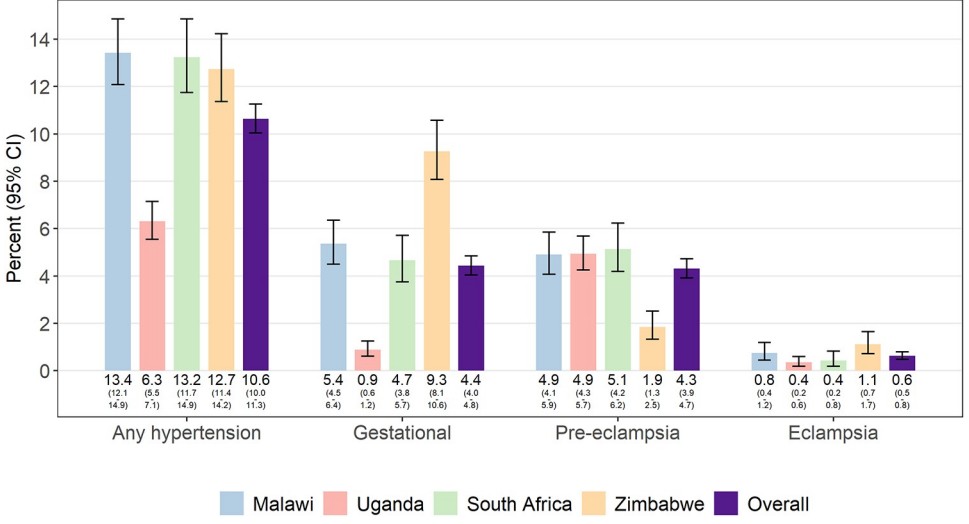

**Fig 2. Frequency of pregnancy complications by country.** Data presented, overall and by site, for n = 10,138 records reviewed. Data on pregnancy-related compilations were missing from <1% of records.

**Table 7. Select characteristics by transfer status from a different facility for delivery.**

| | Not transferred from a different facility | | Transferred from a different facility to current facility for delivery | |
|---|---|---|---|---|
| | N = 5875 | | N = 4485 | |
| Place of delivery | | | | |
| Current health facility | 5418 | (92.2) | 4300 | (95.9) |
| At a different health facility | 260 | (4.4) | 146 | (3.3) |
| At a home (private residence) | 183 | (3.1) | 34 | (0.8) |
| Not documented | 14 | (0.2) | 5 | (0.1) |
| Mode of delivery | | | | |
| Vaginal delivery | 4599 | (78.3) | 2985 | (66.6) |
| Cesarean delivery | 1233 | (21.0) | 1453 | (32.4) |
| Other | 24 | (0.4) | 44 | (1.0) |
| Not documented | 19 | (0.3) | 3 | (0.1) |
| Pregnancy outcomes | | | | |
| Full term live birth | 4874 | (83.0) | 3545 | (79.0) |
| Premature live birth | 643 | (10.9) | 657 | (14.6) |
| Still born/intrauterine fetal demise (> = 20 weeks) | 170 | (2.9) | 230 | (5.1) |
| Outcome not documented | 188 | (3.2) | 53 | (1.2) |
| Deaths | | | | |
| Neonatal death | 77 | (1.4) | 112 | (2.7) |
| Maternal death | 3 | (0.05) | 4 | (0.09) |

Statistics are N (%) for categorical variables.

In this cross-sectional chart review of pregnancy outcomes, neonatal outcomes and pregnancy complications recorded at facilities in four urban centers in eastern and southern Africa, we noted similar frequencies for most outcomes across the different geographic areas. Levels of missing data were extremely low for all study outcomes assessed, providing critical information on pregnancy and neonatal outcomes as well as pregnancy complications for which data from low and middle-income countries (LMICs) are sparse [8]. Estimates from this study provide important background data for ongoing studies assessing the safety of novel HIV prevention products as well as future studies of investigational products evaluated in pregnancy in these urban settings.

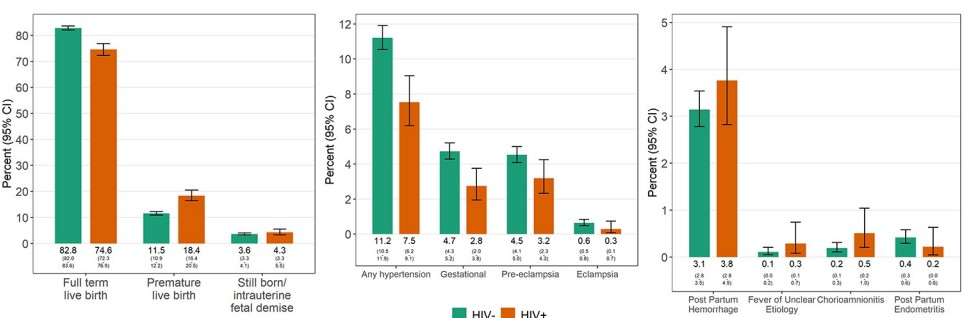

**Fig 3. Pregnancy outcomes and complications by maternal HIV status.** Data presented by maternal HIV status for n = 10,017 individuals where HIV status was documented in the chart.

We observed that site estimates of certain outcomes differed somewhat from national data sources. While national data estimate preterm birth rates of 10.5% for Malawi, 12.4% for South Africa, 6.6% for Uganda and 12.0% for Zimbabwe [15], the prevalence of preterm birth reported in this chart abstraction was slightly higher for Johannesburg, South Africa and Kampala, Uganda. The prevalence of stillbirth estimated using national data was slighter lower than estimates reported in the chart review (national stillbirth estimates for Malawi = 2.2%, South Africa = 1.7%, Uganda = 2.1% and Zimbabwe = 2.1%) [16]. These differences could be a result of underlying differences in the catchment area, differences due to the type of facilities included in the chart review, differences in outcome definition or a combination of these factors. Variation between national estimates and those reported in this analysis highlight the need for local estimates of pregnancy outcomes in order to ensure appropriate comparisons between the frequency of outcomes observed in interventional studies.

While data on pregnancy outcomes and maternal and neonatal mortality are routinely reported as priority health indicators, data for pregnancy complications, which may be more difficult to accurately ascertain, are lacking. A recent systematic review conducted by our group assessed studies conducted in Malawi, South Africa, Uganda and Zimbabwe over the past 20 years that estimated the frequency of pregnancy complications. Few published studies were identified that estimated the prevalence of gestational hypertension (n = 14), pre-eclampsia/eclampsia (n = 9), postpartum endometritis (n = 5) and chorioamnionitis (n = 3) and most studies included a small number of participants or populations that may not be generalizable to the general population of pregnant women (*Lokken et al.* under review at PLOS ONE) [8].

Given reports of higher rates of preterm birth among women living with HIV relative to national preterm birth rates [14,17], we performed a sensitivity analysis comparing the prevalence of study outcomes by maternal HIV status. Consistent with prior studies, we observed a higher frequency of preterm birth among women living HIV compared to those that were not living with HIV. Data were not collected on antiretroviral therapy (ART) use in pregnancy for this review; therefore, we are not able to determine if the frequency of preterm birth differed by ART regimen or compared to HIV-uninfected women. In contrast, we observed lower frequencies of pregnancy-related hypertensive disorders among pregnant women living with HIV compared to HIV-uninfected women and no substantive differences for all other complications. Data are limited on the frequency of pregnancy complication by HIV status [8]. A recent systematic review of studies assessing the impact of ART and the risk of hypertensive disorders in pregnancy reported mixed results with substantive differences in study design and population that likely contributed to variation in the results [18]. High quality, rigorously collected data on pregnancy complications in in low and middle-income countries (LMICs) are urgently needed to better inform care for all women.

In addition to the paucity of published data on pregnancy complications in LMICs, challenges in underreporting of pregnancy complications and variation in outcome definitions may limit comparability across studies even when complications are assessed [19,20]. To improve the quality and comparability of safety data collected as part of maternal immunization studies conducted in LMICs, the Global Alignment of Immunization Safety Assessment in pregnancy project (GAIA) was established in 2015 to develop standardized definitions for pregnancy outcomes, pregnancy complications and neonatal outcomes that can be applied across a range of health care settings [19,21]. In an effort to facilitate comparability, this review utilized GAIA project definitions when possible; however, due to limitations in the data routinely collected in medical charts we were not able to apply those definitions for all pregnancy complications.

This analysis includes several strengths. Rigorous methods for data abstraction were implemented, including the use of standardized outcome definitions and daily quality control

checks to ensure consistency and agreement across abstractors. The level of missing data was low, with less than 2.5% missing data for key outcomes. In addition, data abstractors were able to complete reviews and abstraction for all deliveries that occurred at the facilities during the periods of chart abstraction. Such high levels of completeness and comprehensive data abstraction minimizes bias that could be introduced by incomplete abstraction. However, this analysis should be interpreted in the context of several limitations. Most notably, this review relied on the presence of the diagnosis term in the chart or sufficient clinical information documented in the chart in support of the diagnosis. While we observed high levels of completeness, it is still possible that this review underestimated the frequency of pregnancy complications due to insufficient or variable levels of documentation in the medical chart [22,23]. Suspected congenital anomalies reported in this review only included those that would be identified upon visual examination at the time of birth; anomalies identified later in development or requiring additional diagnostic assessment were not included due to the nature of the study design. Additional data on the reason for transfer to another facility were not available; therefore, the present summaries should be interpreted with caution. Lastly, while our sample size was quite large, it may have been insufficient to capture rare outcomes, such as maternal death, in all settings.

## Conclusions

There is an urgent need and ethical imperative to include pregnant individuals in clinical trials in order to generate safety data in support of biomedical interventions for use during pregnancy, including those for HIV prevention [6]. The data presented in this chart review provide critical information for use by researchers, public health professionals, community members and national regulatory bodies to inform comparisons of outcomes from safety studies of investigational products with those in observed in the general population of pregnant cisgender women in Blantyre, Malawi; Johannesburg, South Africa; Kampala, Uganda; and Chitungwiza and Harare, Zimbabwe.

## Supporting information

**S1 File. MTN-042B data abstraction form.**
(PDF)

## Acknowledgments

### Additional authors from clinical research sites (Alphabetical order)

Luis Gadama, Lonjezo Jemi, Hawah Mbali, Chimwemwe Nkhonjera (College of Medicine-Johns Hopkins Research Project, Blantyre, Malawi); Megan Dempster, Karabo Kungoane, Jean le Roux, Zinhle Tshabalala, Caroline Vika, Sarah-Jane Whitaker (Wits Reproductive Health and HIV Institute, Faculty of Health Sciences, University of the Witwatersrand, South Africa); Naluggwa Abisagi, Faridah Ali, Nakangu Berna, Oloo Keziron Eric, Kemigisa Evelyn, Ekel Irene, Birungi Harriet Mawanda, Atukunda Mediah, Atwebembire Prossy, Mirembe Ritah, Amanya Spincious, Helen Agoile Unzia (Makerere University-Johns Hopkins University Research Collaboration, Kampala, Uganda); Patricia Mae Dhlakama,Vanessa Gatsi, Sibongile Makwenda, Moleen Matimbira, Margaret Mlingo, Mary Mudavanhu, Eneresi Munjoma, Fungayi Murewa, Grecenia Ndhlovu (University of Zimbabwe College of Health Sciences Clinical Trials Research Centre, Harare, Zimbabwe).

## Author Contributions

**Conceptualization:** Jennifer E. Balkus, Lee Fairlie, Bonus Makanani, Ashley Mayo, Tanya Harrell, Jeanna Piper, Katherine E. Bunge.

**Data curation:** Tanya Harrell.

**Formal analysis:** Jennifer E. Balkus, Moni Neradilek.

**Investigation:** Jennifer E. Balkus, Lee Fairlie, Bonus Makanani, Nyaradzo Mgodi, Felix Mhlanga, Clemensia Nakabiito, Katherine E. Bunge.

**Project administration:** Ashley Mayo, Tanya Harrell.

**Writing – original draft:** Jennifer E. Balkus.

**Writing – review & editing:** Jennifer E. Balkus, Moni Neradilek, Lee Fairlie, Bonus Makanani, Nyaradzo Mgodi, Felix Mhlanga, Clemensia Nakabiito, Ashley Mayo, Tanya Harrell, Jeanna Piper, Katherine E. Bunge.

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
