## [Decision Letter · Decision Letter 0]

17 Nov 2020

PONE-D-20-32360

Assessing pregnancy and neonatal outcomes in Malawi, South Africa, Uganda, and Zimbabwe: Results from a systematic chart review

PLOS ONE

Dear Dr. Balkus,

Thank you for submitting your manuscript to PLOS ONE. After careful consideration, we feel that it has merit but does not fully meet PLOS ONE’s publication criteria as it currently stands. Therefore, we invite you to submit a revised version of the manuscript that addresses the points raised during the review process. Please submit your revised manuscript by Jan 01 2021 11:59PM. If you will need more time than this to complete your revisions, please reply to this message or contact the journal office at plosone@plos.org. Please include the following items when submitting your revised manuscript:

We look forward to receiving your revised manuscript.

Kind regards,

Tai-Heng Chen, M.D.

Academic Editor

PLOS ONE

2. Thank you for including your ethics statement: 'Each site received IRB approval and was granted a waiver of informed consent.'

a. Please amend your current ethics statement to include the full names of the ethics committees/institutional review boards that approved your specific study.

4. Please upload a copy of Supporting Information Supplemental Figure 1 which you refer to in your text on page 7.

Reviewers' comments:

Reviewer's Responses to Questions

**Comments to the Author**

1. Is the manuscript technically sound, and do the data support the conclusions?

Reviewer #1: Yes

Reviewer #2: Yes

2. Has the statistical analysis been performed appropriately and rigorously? 

Reviewer #1: Yes

Reviewer #2: Yes

3. Have the authors made all data underlying the findings in their manuscript fully available?

Reviewer #1: Yes

Reviewer #2: Yes

4. Is the manuscript presented in an intelligible fashion and written in standard English?

Reviewer #1: Yes

Reviewer #2: Yes

5. Review Comments to the Author

Reviewer #1: This is a multi-site, cross-sectional chart review to estimate the frequency of pregnancy outcomes, pregnancy complications and neonatal outcomes at four urban centers across four African countries (Blantyre, Malawi; Johannesburg, South Africa; Kampala, Uganda; and Chitungwiza and Harare). These data could provide important background information of pregnancy outcomes in Southeastern African countries, which may have public health value.

In the METHODS, “Initially, each site selected one or two facilities for data abstraction, representing a balance of primary and tertiary care facilities to ensure that the data abstracted would be representative of outcomes experienced by the general population in the catchment area.”

Q. All selected facilities in this study seemed to be tertiary hospitals and located in urban area rather than primary care hospitals in the rural areas.

To make up for this potential shortcoming, I would suggest using the “Transferred for delivery from a different facility (43.1%)" as a proxy, because they are most likely to be transferred from rural hospitals and could represent women in rural areas to some extent. I hope that the authors can further compare the difference between the two groups of "transferred" and "non-transferred" women. The results of the analysis may increase the richness of this paper.

Do all HIV (+) women take ARV prescriptions? Although there is no ARV regimen information for every woman, in principle, the same country will use the same regimen. Please sort out the ARV regimen of all countries for reference. And try to analyze the correlation between these regimens and complication rates.

Reviewer #2: The purpose of this systematic chart review is to make available pregnancy outcome data from multiple African sites to help in assessment of adverse events when new devices and drugs are evaluated during pregnancy and immediate postpartum. Overall, this is a simple analysis. Few comments below:

1. The investigators mention that that they did a review of the literature as part of the study. Would be a good complementary material to summarize those findings in a table showing frequency of the adverse outcome in African countries - preferably stratified by HIV infection status and antiretroviral use - showing time of study conduct and African region.

2. The investigators in more than one place mention that they are collecting this data to compare with outcomes from an on-going clinical trial ..... this appears misleading since no description of that study or conduct of that study is part of the writeup. Would be fine to state as part of the overall intent only.

3. The aims that appear at the beginning of Methods section should be moved to end of Introduction section.

4. In Methods provide definition of pregnancy outcomes used. Were these based on newborn weight and gestational age measurement checks ---- or based on a recorded diagnosis only (in this case it is a major limitation). If gestational age and birth weight measurements were abstracted (available) some validity checks - even on a sample - would be helpful.

5. Missing data seem extremely low: Is this a bias with the abstraction process -- i.e., forms with extensive missing data are not included altogether because there is nothing there to abstract? How many at each site met this potential abstraction problem?

6. Tables and other analyses are fine.

6. PLOS authors have the option to publish the peer review history of their article (what does this mean?). If published, this will include your full peer review and any attached files.

Reviewer #1: **Yes: **Solomon Chih-Cheng Chen

Reviewer #2: No

---

## [Author Response · Author response to Decision Letter 0]

17 Dec 2020

Reviewer #1: 

This is a multi-site, cross-sectional chart review to estimate the frequency of pregnancy outcomes, pregnancy complications and neonatal outcomes at four urban centers across four African countries (Blantyre, Malawi; Johannesburg, South Africa; Kampala, Uganda; and Chitungwiza and Harare). These data could provide important background information of pregnancy outcomes in Southeastern African countries, which may have public health value.

1. In the METHODS, “Initially, each site selected one or two facilities for data abstraction, representing a balance of primary and tertiary care facilities to ensure that the data abstracted would be representative of outcomes experienced by the general population in the catchment area.” All selected facilities in this study seemed to be tertiary hospitals and located in urban area rather than primary care hospitals in the rural areas. To make up for this potential shortcoming, I would suggest using the “Transferred for delivery from a different facility (43.1%)" as a proxy, because they are most likely to be transferred from rural hospitals and could represent women in rural areas to some extent. I hope that the authors can further compare the difference between the two groups of "transferred" and "non-transferred" women. The results of the analysis may increase the richness of this paper.

We appreciate the opportunity to clarify the selection of facilities for data abstraction. Since the primary goal of this work was to describe the frequency of pregnancy outcomes, pregnancy complications and neonatal outcomes among individuals delivering in specified catchment areas and compare those with estimates from an ongoing clinical trial evaluating the safety of the two biomedical interventions for HIV prevention in pregnancy, we selected delivery facilities from within those catchment areas. We do indeed have a balance of primary and tertiary care facilities from the catchment area, which may or may not include peri-urban and rural areas. Since we did not collect residential information or the location where the individual was transferred from, we are not able to provide further insight into differences in outcomes by geographic location. 

2. Do all HIV (+) women take ARV prescriptions? Although there is no ARV regimen information for every woman, in principle, the same country will use the same regimen. Please sort out the ARV regimen of all countries for reference. And try to analyze the correlation between these regimens and complication rates.

The impact of antiretroviral therapy (ART) on pregnancy outcomes is an important research question; however, given the goal of this chart review, we did not collect data on ART use at the time of delivery. Malawi, South Africa, Uganda and Zimbabwe each have national guidance for ART use which is based on WHO recommendations and regimens may vary by country and whether an individual is on a first-or second-line regimen. Since we can’t say for certain which regimens, if any, women were using at the time of delivery, we are not able to conduct any additional analysis among women who are living with HIV. This was noted as a limitation in our discussion.

Reviewer #2: 

The purpose of this systematic chart review is to make available pregnancy outcome data from multiple African sites to help in assessment of adverse events when new devices and drugs are evaluated during pregnancy and immediate postpartum. Overall, this is a simple analysis. Few comments below:

3. The investigators mention that that they did a review of the literature as part of the study. Would be a good complementary material to summarize those findings in a table showing frequency of the adverse outcome in African countries - preferably stratified by HIV infection status and antiretroviral use - showing time of study conduct and African region.

The findings from the systematic review have been submitted to PLoS One for publication and include a detailed review of 10 different outcomes (Dr. Erica Lokken is the first author and the manuscript is currently under review). The review is incredibly detailed an includes analysis by region as well as by HIV status. Given the detail provided in the review, we hope to be able to cite the findings in this manuscript rather than duplicate what is presented in the review. 

4. The investigators in more than one place mention that they are collecting this data to compare with outcomes from an on-going clinical trial ..... this appears misleading since no description of that study or conduct of that study is part of the writeup. Would be fine to state as part of the overall intent only.

We would like to highlight that a description of the DELIVER study is provided in the first paragraph of the introduction along with a reference linking to the study protocol and the clinicaltrials.gov registration number. Because the DELIVER study is ongoing and the focus of this manuscript was on reporting our findings from the chart review, which may have comparative benefit above and beyond the DELIVER study, we felt is was appropriate to only include a brief over view of the DELIVER study. The final sentence in the review comment leads us to believe that the review agrees with this approach. 

5. The aims that appear at the beginning of Methods section should be moved to end of Introduction section.

Thank you for this suggestion, we have moved the sentence that describes the aim of the study to the end of the introduction. 

6. In Methods provide definition of pregnancy outcomes used. Were these based on newborn weight and gestational age measurement checks ---- or based on a recorded diagnosis only (in this case it is a major limitation). If gestational age and birth weight measurements were abstracted (available) some validity checks - even on a sample - would be helpful.

The gestational age at delivery was assessed using a combindation of data available in the medical chart including, date of last menstrual period (LMP), estimated date of delivery based on ultrasound and fundal height. We have added this information to the methods. 

7. Missing data seem extremely low: Is this a bias with the abstraction process -- i.e., forms with extensive missing data are not included altogether because there is nothing there to abstract? How many at each site met this potential abstraction problem?

We appreciate the opportunity to clarify the volume of missing data. No charts were excluded based on level of missing data. For pregnancy outcomes, pregnancy complications and neonatal outcomes, data abstractors were asked to determine the outcome/complication based on the presence of the diagnosis term in the chart or the presumption of the diagnosis based on other available data in the chart that supported the diagnosis based on our a priori definitions (see Table 2). This approach allowed for a higher-than-expected complete data, with < 2% data missing only neonatal death congenital anomalies, and hypertension (missing data is noted in the text and footnotes of the tables). HIV infection status was missing from ~4% of records and those without complete data were excluded from the sensitivity analysis stratified by HIV status. In the discussion, we noted that while this approach allowed us to determine an outcome for the vast majority if records, it is possible that we underestimated the frequency of pregnancy complications due to insufficient or variable levels of documentation in the medical chart. 

8. Tables and other analyses are fine.

Thank you for this note.

---

## [Decision Letter · Decision Letter 1]

8 Jan 2021

PONE-D-20-32360R1

Assessing pregnancy and neonatal outcomes in Malawi, South Africa, Uganda, and Zimbabwe: Results from a systematic chart review

PLOS ONE

Dear Dr. Balkus,

Thank you for submitting your manuscript to PLOS ONE. After careful consideration, we feel that it has merit but does not fully meet PLOS ONE’s publication criteria as it currently stands. Therefore, we invite you to submit a revised version of the manuscript that addresses the points raised during the review process.

We look forward to receiving your revised manuscript.

Kind regards,

Tai-Heng Chen, M.D.

Academic Editor

PLOS ONE

Reviewers' comments:

Reviewer's Responses to Questions

**Comments to the Author**

1. If the authors have adequately addressed your comments raised in a previous round of review and you feel that this manuscript is now acceptable for publication, you may indicate that here to bypass the “Comments to the Author” section, enter your conflict of interest statement in the “Confidential to Editor” section, and submit your "Accept" recommendation.

Reviewer #1: (No Response)

Reviewer #2: All comments have been addressed

2. Is the manuscript technically sound, and do the data support the conclusions?

Reviewer #1: Partly

Reviewer #2: Yes

3. Has the statistical analysis been performed appropriately and rigorously? 

Reviewer #1: No

Reviewer #2: Yes

4. Have the authors made all data underlying the findings in their manuscript fully available?

Reviewer #1: No

Reviewer #2: Yes

5. Is the manuscript presented in an intelligible fashion and written in standard English?

Reviewer #1: No

Reviewer #2: Yes

6. Review Comments to the Author

Reviewer #1: I am not satisfied with the author's Response because they did not respond to my suggestions appropriately.

For example, these 43% cases "Transferred from different facility" are already available information, but the authors refuse to analyze it further.

Your Response: ”We do indeed have a balance of primary and tertiary care facilities from the catchment area, which may or may not include peri-urban and rural areas.”

Please specify how the authors did indeed obtain "a balance of primary and tertiary care facilities"? Please list in detail which medical institutions are PRIMARY and how many people are there? Which medical facilities are TERTIARY and how many people are there?

Reviewer #2: The authors should cite the pending systematic review. The analyses will help interpretation of data from the on-going trial as well as data addressing use and safety of experimental devices during pregnancy in African women.

7. PLOS authors have the option to publish the peer review history of their article (what does this mean?). If published, this will include your full peer review and any attached files.

Reviewer #1: No

Reviewer #2: No

---

## [Author Response · Author response to Decision Letter 1]

21 Jan 2021

Dear Dr. Chen, 

Thank you for the additional reviewer comments for our manuscript titled “Assessing pregnancy and neonatal outcomes in Malawi, South Africa, Uganda, and Zimbabwe: Results from a systematic chart review.” Please see our revisions in the track changes version of the manuscript, as well as our responses to the additioanl reviewers’ comments in the following pages. The reviewers’ comments are written in plain text and our responses are provided in italics. We recognize that reviewer #1 had two additional comments based on our prior responses. We hope that these most recent modifications will be acceptable to the reviewer and the editors at PLOS ONE. Thank you again for considering our submission and we look forward to hearing from you. 

Sincerely,

Jennifer Balkus, PhD, MPH

Assistant Professor

Department of Epidemiology

University of Washington School of Public Health

Reviewer #1: 

1. I am not satisfied with the author's Response because they did not respond to my suggestions appropriately. For example, these 43% cases "Transferred from different facility" are already available information, but the authors refuse to analyze it further. 

We appreciate the reviewer’s comment and the opportunity to provide further information on this topic. While we abstracted data on whether an individual was transferred from a different facility, we did not collect any additional information on the reasons why a transfer occurred. The settings where data abstraction was conducted, transfers can occur for a variety of reasons, some of which are not directly linked to the need for higher level care. The maternity care system in each of these settings is based upon a group of primary care facilities that feed into the tertiary care facility for their health region, thus all women in the catchment area would be expected to deliver in one of these two locations. Pregnant individuals typically present to the primary care facility closest to their home but are transferred to the tertiary facility if they are either known to be high risk upon presentation from their antenatal care (e.g., prior cesarean delivery) or develop high risk characteristics during labor, delivery or postpartum (e.g., preeclampsia). However, both low and high-risk individuals may also present directly to the tertiary facility, or in some cases, the tertiary facility also serves as the primary facility for the catchment area around the research clinic (such is the case for Uganda). This is consistent with the fact that while ~43% of records indicated a transfer for care, ~94% of deliveries occurred at the facility where data abstraction was performed. 

Given that data were not available regarding the reason an individual was transferred, we were concerned about the interpretation of such data in the absence of additional context, thus our reluctance to perform additional analysis stratified by this variable. To address the reviewer’s comment above we have performed an additional analysis describing pregnancy outcomes and maternal and neonatal deaths among those that were transferred from a different facility and those that were not. Data are provided in the table below. (Please see attached letter for formatted table as it does not display properly here). 

Select characteristics by the mother’s transfer for delivery from a different facility

 Not transferred from a different facility Transferred from a different facility to current facility Overall

 N=5875 N=4485 N=10426

Place of delivery 

 Current health facility 5418 (92.2) 4300 (95.9) 9779 (93.8)

 At a different health facility 260 (4.4) 146 (3.3) 408 (3.9)

 At a home (private residence) 183 (3.1) 34 (0.8) 218 (2.1)

 Not documented 14 (0.2) 5 (0.1) 21 (0.2)

Mode of delivery 

 Vaginal delivery 4599 (78.3) 2985 (66.6) 7622 (73.1)

 Cesarean delivery 1233 (21.0) 1453 (32.4) 2710 (26.0)

 Other 24 (0.4) 44 (1.0) 69 (0.7)

 Not documented 19 (0.3) 3 (0.1) 25 (0.2)

Pregnancy outcomes 

 Full term live birth 4874 (83.0) 3545 (79.0) 8448 (81.0)

 Premature live birth 643 (10.9) 657 (14.6) 1319 (12.7)

 Still born/intrauterine fetal demise (>= 20 weeks) 170 (2.9) 230 (5.1) 413 (4.0)

 Outcome not documented 188 (3.2) 53 (1.2) 246 (2.4)

Deaths 

 Neonatal death 77 (1.4) 112 (2.7) 192 (2.0)

 Maternal death 3 (0.05) 4 (0.09) 7 (0.07)

 Statistics are N (%) for categorical variables and mean±SD (range) for continuous variables.

The estimates listed by transfer status are similar to the overall estimates and the range of estimates by site (see Table 4 for estimates by country). Since the reason for transfer was not recorded, we continue to have some concerns about the interpretation of these data. However, we have included the above table in the revised manuscript with additional text throughout the manuscript as appropriate.

2. Your Response: ”We do indeed have a balance of primary and tertiary care facilities from the catchment area, which may or may not include peri-urban and rural areas.” Please specify how the authors did indeed obtain "a balance of primary and tertiary care facilities"? Please list in detail which medical institutions are PRIMARY and how many people are there? Which medical facilities are TERTIARY and how many people are there?

To address this comment we have added a column to Table 1 to indicate which facilities are considered primary and which are considered tertiary. Our local collaborators identified the primary and tertiary facility in their region that potential participants would most likely attend for delivery. For example, our Johannesburg site included the Shandukani Maternal and Obstetrics Unit, which is a primary unit located in the same building as the research clinic, and the Charlotte Maxeke Johannesburg Academic Hospital, which is the tertiary facility that serves residents of the area. One exception to this was the Harare site that was forced to switch facilities due to industrial actions (protests) in the area that resulted in patients, including those seeking obstetric services, needing to seek care at other clinics and hospitals (as described in the methods section of the manuscript). As described in the updated manuscript, the facility used in Uganda provides both primary and tertiary care for the site’s catchment area so a single facility was utilized there.

Regarding the second question about the number of individuals at each facility, all individuals delivering at the designated facility or admitted to the facility for postpartum care within seven days of a delivery elsewhere (home, health clinic, etc.) were included in the review. The number of records evaluated during the periods of review at each facility is displayed in Table 1. 

Reviewer #2: 

I have reviewed the responses and the revised manuscript. I approve the responses with one condition: The authors should cite the systematic review that is pending review.

1. Since we submitted our previous letter (15 December 2020), we received an invitation from PLOS One to revise and resubmit the systematic review. We have modified the current text to indicate the first author of the systematic review and that is is under review at PLOS One, in case the current manuscript is published before the systematic review.

---

## [Decision Letter · Decision Letter 2]

16 Feb 2021

PONE-D-20-32360R2

Assessing pregnancy and neonatal outcomes in Malawi, South Africa, Uganda, and Zimbabwe: Results from a systematic chart review

PLOS ONE

Dear Dr. Balkus,

Thank you for submitting your manuscript to PLOS ONE. After careful consideration, we feel that it has merit but does not fully meet PLOS ONE’s publication criteria as it currently stands. Therefore, we invite you to submit a revised version of the manuscript that addresses the points raised during the review process.

We look forward to receiving your revised manuscript.

Kind regards,

Tai-Heng Chen, M.D.

Academic Editor

PLOS ONE

Reviewers' comments:

Reviewer's Responses to Questions

**Comments to the Author**

1. If the authors have adequately addressed your comments raised in a previous round of review and you feel that this manuscript is now acceptable for publication, you may indicate that here to bypass the “Comments to the Author” section, enter your conflict of interest statement in the “Confidential to Editor” section, and submit your "Accept" recommendation.

Reviewer #1: All comments have been addressed

Reviewer #3: All comments have been addressed

Reviewer #4: (No Response)

2. Is the manuscript technically sound, and do the data support the conclusions?

Reviewer #1: Yes

Reviewer #3: Yes

Reviewer #4: Yes

3. Has the statistical analysis been performed appropriately and rigorously? 

Reviewer #1: Yes

Reviewer #3: Yes

Reviewer #4: Yes

4. Have the authors made all data underlying the findings in their manuscript fully available?

Reviewer #1: Yes

Reviewer #3: Yes

Reviewer #4: Yes

5. Is the manuscript presented in an intelligible fashion and written in standard English?

Reviewer #1: Yes

Reviewer #3: Yes

Reviewer #4: Yes

6. Review Comments to the Author

Reviewer #1: I highly appreciate authors' efforts to do further analysis and interpretation for the “transferred cases”. In addition to the cesarean delivery (32.4% versus 21.0%) and preterm birth (14.6% versus 10.9%), the almost twice death rates (2.7% versus 1.4% in neonatal death, and 0.09% versus 0.05% in maternal deaths) may be significant too. The above results really highlight the importance of “transfer” issue and cannot be ignored.

Reviewer #3: Many thanks for the opportunity to review this interesting original article. This article was a systematic chart review of pregnancy outcomes, pregnancy complications and neonatal outcomes in Malawi, South Africa, Uganda and Zimbabwe. This was an important paper because it could clarify the pregnancy and neonatal outcomes of these specified urban settings in Africa. Also, it may be clarified the effect of HIV prevention in these area by comparing ongoing study (DELIVER study).

Reviewer #4: Balkus et al analyzed the pregnancy outcomes, neonatal outcomes and pregnancy complications in eastern and southern Africa. To cover various maternal backgrounds and perinatal-care factors, the authors included both primary and tertiary facilities in each country. The authors summarized the frequency of pregnancy characteristics and outcomes. In addition, the authors compared the pregnancy outcomes between the HIV-positive and HIV-negative groups. This large population-based study enhances our understanding of maternal and neonatal outcomes in the area concerned. As the authors claimed, these results can be further used for evaluating the safety of biomedical interventions for HIV prevention during pregnancy. I believe that the data collection and analysis were performed properly. The conclusions that the authors draw from the data are fine. Furthermore, the manuscript has been revised well through the revisions. Therefore, I think this manuscript is now acceptable for publication.

7. PLOS authors have the option to publish the peer review history of their article (what does this mean?). If published, this will include your full peer review and any attached files.

Reviewer #1: **Yes: **Solomon Chih-cheng Chen

Reviewer #3: No

Reviewer #4: No

---

## [Author Response · Author response to Decision Letter 2]

23 Feb 2021

Dear Dr. Chen, 

Thank you for the additional reviewer comments from this latest round of review for our manuscript titled “Assessing pregnancy and neonatal outcomes in Malawi, South Africa, Uganda, and Zimbabwe: Results from a systematic chart review.” On 16 February 2021, we received your email that included reviewer comments from a third round of review of our paper. We were pleased to note that all reviewers stated that their comments from the previous revision that was submitted on 20 January 2021 were addressed. However, in your email you ask us to address an outstanding comment from reviewer #1. In the text below, we clarify that there are actually no outstanding comments from reviewer #1 as they were all addressed in our responses that were submitted on 20 January 2021. In our second round of review, reviewer #1 had continued concerns about the fact that we did not include any additional exploratory analysis by hospital transfer status. Their comment from the second review related to this concern is provided below in italics:

Reviewer #1 comment from second review: 

1. I am not satisfied with the author's Response because they did not respond to my suggestions appropriately. For example, these 43% cases "Transferred from different facility" are already available information, but the authors refuse to analyze it further. 

In response to this comment, we provided an extensive explanation for our initial approach in the response letter from 20 January 2021 and included a new analysis and new table in the paper (Table 3). In the email received on 16 February 2021, reviewer #1 noted "I highly appreciate authors' efforts to do further analysis and interpretation for the “transferred cases”. In addition to the cesarean delivery (32.4% versus 21.0%) and preterm birth (14.6% versus 10.9%), the almost twice death rates (2.7% versus 1.4% in neonatal death, and 0.09% versus 0.05% in maternal deaths) may be significant too. The above results really highlight the importance of “transfer” issue and cannot be ignored." This comment indicates that reviewer #1 supports the addition of the new Table 3, as it provides important summaries regarding pregnancy outcomes by transfer status which reviewer #1 lists in their comment above as affirmation for Table 3. Per the comments included in the 16 February 2021 email, reviewer #1 did not have any additional comments, did not request or suggest any additional analyses and did indicate in that all comments had been addressed. 

We note that the other reviewers also indicated that all comments had been addressed and reviewer #4 states “Furthermore, the manuscript has been revised well through the revisions. Therefore, I think this manuscript is now acceptable for publication.” Given the positive comments from all reviewers and the fact that all reviewers, including reviewer #1, indicated that all comments have been addressed in our last revision, we have not made any modifications to the manuscript. We hope that this letter clarifies how we have already addressed prior reviewer comments and the editorial team will now find the manuscript suitable for publication. 

On behalf of my co-authors, thank you again for considering our submission and we look forward to hearing from you. 

Sincerely,

Jennifer Balkus, PhD, MPH

---

## [Decision Letter · Decision Letter 3]

26 Feb 2021

Assessing pregnancy and neonatal outcomes in Malawi, South Africa, Uganda, and Zimbabwe: Results from a systematic chart review

PONE-D-20-32360R3

Dear Dr. Balkus,

We’re pleased to inform you that your manuscript has been judged scientifically suitable for publication and will be formally accepted for publication once it meets all outstanding technical requirements.

Kind regards,

Tai-Heng Chen, M.D.

Academic Editor

PLOS ONE

Reviewers' comments:

Reviewer's Responses to Questions

**Comments to the Author**

1. If the authors have adequately addressed your comments raised in a previous round of review and you feel that this manuscript is now acceptable for publication, you may indicate that here to bypass the “Comments to the Author” section, enter your conflict of interest statement in the “Confidential to Editor” section, and submit your "Accept" recommendation.

Reviewer #1: All comments have been addressed

2. Is the manuscript technically sound, and do the data support the conclusions?

Reviewer #1: Yes

3. Has the statistical analysis been performed appropriately and rigorously? 

Reviewer #1: Yes

4. Have the authors made all data underlying the findings in their manuscript fully available?

Reviewer #1: Yes

5. Is the manuscript presented in an intelligible fashion and written in standard English?

Reviewer #1: Yes

6. Review Comments to the Author

Reviewer #1: I really appreciate your efforts to respond to my comments. The current form is satisfactory to me. I hope this article can help the world learn more about maternal and child health in Africa. Thank you a lot.

7. PLOS authors have the option to publish the peer review history of their article (what does this mean?). If published, this will include your full peer review and any attached files.

Reviewer #1: **Yes: **Solomon Chih-Cheng Chen

---

## [Editor Report · Acceptance letter]

3 Mar 2021

PONE-D-20-32360R3 

Assessing pregnancy and neonatal outcomes in Malawi, South Africa, Uganda, and Zimbabwe: Results from a systematic chart review 

Dear Dr. Balkus:

I'm pleased to inform you that your manuscript has been deemed suitable for publication in PLOS ONE. Congratulations! Your manuscript is now with our production department. 

Kind regards, 

on behalf of

Dr. Tai-Heng Chen 

Academic Editor

PLOS ONE